# Computational prediction of new auxetic materials

John Dagdelen[1], Joseph Montoya[1], Maarten de Jong[1,2] & Kristin Persson[1,3]

Auxetics comprise a rare family of materials that manifest negative Poisson's ratio, which causes an expansion instead of contraction under tension. Most known homogeneously auxetic materials are porous foams or artificial macrostructures and there are few examples of inorganic materials that exhibit this behavior as polycrystalline solids. It is now possible to accelerate the discovery of materials with target properties, such as auxetics, using high-throughput computations, open databases, and efficient search algorithms. Candidates exhibiting features correlating with auxetic behavior were chosen from the set of more than 67 000 materials in the Materials Project database. Poisson's ratios were derived from the calculated elastic tensor of each material in this reduced set of compounds. We report that this strategy results in the prediction of three previously unidentified homogeneously auxetic materials as well as a number of compounds with a near-zero homogeneous Poisson's ratio, which are here denoted "anepirretic materials".

---

[1] Lawrence Berkeley National Laboratory, 1 Cyclotron Rd, Berkeley, CA 94720, USA. [2] SpaceX, 1 Rocket Road, Hawthorne, CA 90250, USA. [3] University of California, Berkeley, 210 Hearst Memorial Mining Building, Berkeley, CA 94720, USA. Correspondence and requests for materials should be addressed to K.P. (email: kristinpersson@berkeley.edu)

A material's Poisson's ratio reflects the deformation of its cross-section in response to an orthogonal tensile strain. The term "auxetic" defines a class of exotic materials that exhibit a negative Poisson's ratio, which causes a counter-intuitive expansion under tension rather than contraction[1]. These materials have been shown to possess enhanced hardness and toughness, as well as absorb vibrations and sound better than their non-auxetic counterparts[2, 3]. As a result, the atypical elastic behavior of auxetic materials is enabling advancements in a broad range of technologies such as impact-resistant composites, extremely precise sensors, tougher ceramics, and high-performance armor[2–5].

Auxetic behavior is generally believed to originate at the structural level. Materials with anisotropic mechanical behavior, such as crystalline materials, exhibit orientation-dependent directional Poisson's ratios, $\nu_{ij}$ (where $i$ and $j$ are the directions of the applied tensile strain and the resultant transverse strain). For polycrystalline solids, which are macroscopically isotropic, these directional values average to a "homogeneous" value, $\mu^2$. A number of crystalline materials have been found to exhibit negative Poisson's ratios in certain directions associated with specific features of their crystal structures[6–9]. However, nearly all of the currently known homogeneously auxetic materials are porous foams or purposefully designed hinged meta-materials with open, re-entrant structures[3, 10, 11]. The most well-known example of a crystalline material with a negative homogeneous Poisson's ratio, $\alpha$-cristobalite ($SiO_2$), was discovered via laser Brillouin spectroscopy in 1992 by Yeganeh-Haeri et al.[12, 13]. They hypothesized that the material's unusual elastic behavior originates from the rotation of rigid $SiO_4$ tetrahedra. Indeed, the prevailing theory for auxetic behavior in $\alpha$-cristobalite and other inorganic materials relies on rigid unit modes (RUM's) in which connected polyhedra rotate without deforming (see Fig. 1)[2, 3, 14–17]. Interestingly, the unique deformation behavior of this class of materials may also induce specific phase transformations. For example, at high temperature, $\alpha$-cristobalite transforms into $\beta$-cristobalite where the structural transformation path is characterized by a static rotation of the silica tetrahedra[18, 19].

Some zeolites with RUM features have been predicted as auxetic using semi-empirical molecular force field models[20, 21]; however, subsequent experiments were only able to verify negative Poisson's ratio for certain directions, not homogeneous auxetic behavior. Goldstein et al. report negative average Poisson's ratios for a small number of tetragonal and cubic materials calculated from previously published elastic constants[22, 23]. These include $C_{16} FeGe_2$, elemental Ba, and solid solutions based on rocksalt-structured SmS. However, we note that other reference work cites the Poisson's ratio of Ba as 0.28[24] and the calculated Poisson's ratio of $C_{16} FeGe_2$ is obtained as 0.24[25]. Previous studies have also found unusual Poisson's ratios to correlate with elastic anisotropy[26, 27]. In fact, auxetic behavior

in certain directions is not uncommon for materials with some degree of anisotropy, such as cubic metals[6] and layered materials[2, 26, 28]. However, it is worth emphasizing that the existence of auxetic directions alone is not enough to render a material's bulk, average Poisson's ratio negative. Finally, we note that—with the possible exceptions listed above—no new homogeneously auxetic crystalline materials have been discovered since $\alpha$-cristobalite $SiO_2$.

In this work, we utilize a screening process for auxetic materials that employs open-source tools and algorithms to efficiently and systematically find promising candidates. Several open materials databases with structural information are available today[29, 30]; for this study, we use the Materials Project[31], which provides access to structure information for more than 67 000 materials and open-source structure manipulation software[32]. This process yields 38 candidate materials, which are subsequently investigated via first-principles density functional theory (DFT) calculations of their full elastic tensors. Of the 38 candidates, 7 new compounds are identified to exhibit negative homogeneous Poisson's ratios in addition to the known auxetic $\alpha$-cristobalite. The most notable of these predictions is a high temperature aluminum orthophosphate polymorph, HT-$AlPO_4$. The other new materials predicted to exhibit auxetic behavior are four hypothetical $SiO_2$ polymorphs and two hypothetical metal vanadate structures. Additionally, we report near-zero homogeneous Poisson's ratios for a number of materials in various chemical systems, which we denote "anepirretic materials" after the Greek word for "unaffected."

## Results

**Screening process**. To date, the Materials Project[31] contains data for over 67,000 distinct inorganic crystalline materials where of over 4000 have an associated calculated elastic tensor. Benchmarks of the Materials Project elastic tensor workflow suggest that it reproduces elastic-tensor derived bulk and shear moduli well; within 15% of experimentally measured values. However, the workflow requires computationally demanding

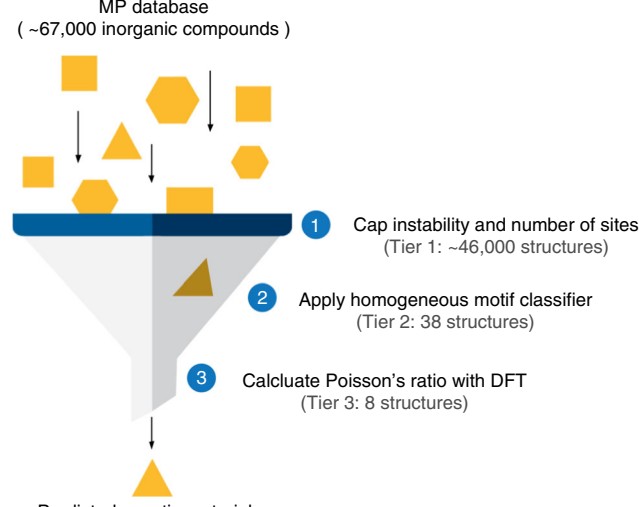

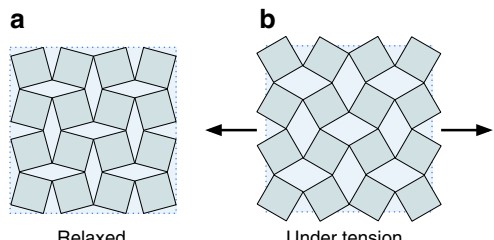

**Fig. 1** The "rotating squares" model for auxetic behavior. **a** The relaxed structure under no strain and **b** the structure under uniaxial tension showing positive strain along both axes

**Fig 2** Overview of the screening process. Tier I filtered on formation energy and number of sites, thereby resulting in a reduction from over 67,000 to ~46,000 compounds. Tier II applied the homologous structure classifier which resulted in a reduction from 46,000 to 38 compounds, including $\alpha$-cristobalite $SiO_2$. In the final step, Tier III, DFT calculations of the full elastic tensors identified 8 structures ($\alpha$-cristobalite included) predicted to exhibit negative homogeneous Poisson's ratios

**Table 1 Predicted directional and homogeneous Poisson's ratios for a selection of Tier 2 materials**

| Material | Structure | Space group | Prev. synthesized? | Directional Poisson | | Homo. Poisson ($\mu$) | Class |
|---|---|---|---|---|---|---|---|
| | | | | ($\nu_{min}$) | ($\nu_{max}$) | | |
| $SiO_2$ ($\alpha C$) | Tetragonal | $P4_12_12$ [92] | Yes[12] | −0.580 | −0.030 | −0.29 | Auxetic |
| HT-$AlPO_4$ | Orthorhombic | $C222_1$ [20] | Yes[18] | −0.582 | −0.043 | −0.28 | Auxetic |
| $SiO_2$ (a) | Tetragonal | $P4_32_12$ [96] | No | −0.551 | −0.029 | −0.27 | Auxetic |
| $SiO_2$ (b) | Orthorhombic | $P2_12_12_1$ [19] | No | −0.518 | 0.027 | −0.20 | Auxetic |
| $SiO_2$ (c) | Orthorhombic | $P2_1nb$ [33] | No | −0.478 | 0.168 | −0.05 | Anepirretic |
| $FeV_3O_8$ | Triclinic | $P1$ [1] | No | −0.379 | 0.168 | −0.05 | Anepirretic |
| $CoV_3O_8$ | Triclinic | $P1$ [1] | No | −0.452 | 0.293 | −0.04 | Anepirretic |
| $SiO_2$ (d) | Monoclinic | $C2/c$ [15] | No | −0.651 | 0.403 | −0.01 | Anepirretic |
| $FePO_4$ | Orthorhombic | $Pn2_1a$ [33] | No | −0.328 | 0.269 | 0.01 | Anepirretic |
| $SiO_2$ ($\beta C$) | Orthorhombic | $I\overline{4}2d$ [122] | Yes[43] | −0.102 | 0.027 | 0.05 | Anepirretic |
| $BVO_4$ | Tetragonal | $I\overline{4}$ [82] | No | −0.502 | 0.392 | 0.06 | Anepirretic |
| $GaPO_4$ | Orthorhombic | $C222_1$ [20] | Yes[42] | −0.343 | 0.548 | 0.07 | Anepirretic |
| $MnV_3O_8$ | Triclinic | $P1$ [1] | No | −0.338 | 0.492 | 0.09 | Anepirretic |
| $GeO_2$ | Tetragonal | $P4_12_12$ [92] | Yes[51] | −0.301 | 0.448 | $0.10^+$ | Meiotic |

input parameters to ensure this level of accuracy[25]. It is currently projected to take years to calculate the elastic tensor for all known crystalline compounds using available supercomputing resources. Hence, approaches which prioritize certain compounds based on structure-property descriptors are useful to accelerate the rate of discovery. For example, predictive statistical learning methods have been used to estimate bulk and shear moduli of materials in the MP database, illustrating that existing DFT-based data on elasticity and structure can be leveraged to guide the search for specific elastic behavior[33]. In this work, to expedite the discovery process with respect to Poisson's ratio, materials were screened though a set of criteria designed to reduce the number of elasticity calculations required to identify novel auxetics. In Tier I (see Fig. 2), we first narrowed our candidate pool by applying a stability criterion, retaining only materials for which the formation energy is less than 100 meV per atom above the convex hull. This removes materials from consideration that may be challenging to synthesize[34]. From this set, computationally tractable materials with fewer than 50 sites per unit cell were retained, resulting in roughly 46,000 structures (see Fig. 2, Tier I). We note that this constraint necessarily excludes many zeolites and porous materials, which may exhibit negative Poisson's ratios[8, 20]. However, auxetic behavior in those materials may be due to structural motifs at a larger scale than those targeted in our search[16].

In Tier II, we used a homologous structure motif to identify materials likely to exhibit auxetic behavior using $\alpha$-cristobalite as a structural archetype. The advantage of homologous structure matching over methods using select geometric descriptors is that information with seemingly inconspicuous significance is inherently included in the crystallographic description of the structure. This is especially relevant for inorganic crystalline auxetic materials because so few positive examples exist. To implement the structural screening, we utilized the pymatgen structure matcher[32] to construct a structure similarity classifier which compared the roughly 46,000 materials that passed the first tier to $\alpha$-cristobalite's corner-connected tetrahedral motif. A species-agnostic algorithm was used to initialize the structure matcher in order to expand the scope of the search beyond binary chemical systems.

The result of the overall screening process was a reduction in the number of candidate materials for full DFT elasticity calculations by three orders of magnitude, from more than 67 000 materials to 38. We note that this set included the known auxetic $\alpha$-cristobalite $SiO_2$. The Poisson's ratios of the candidate compounds were subsequently calculated using the Materials

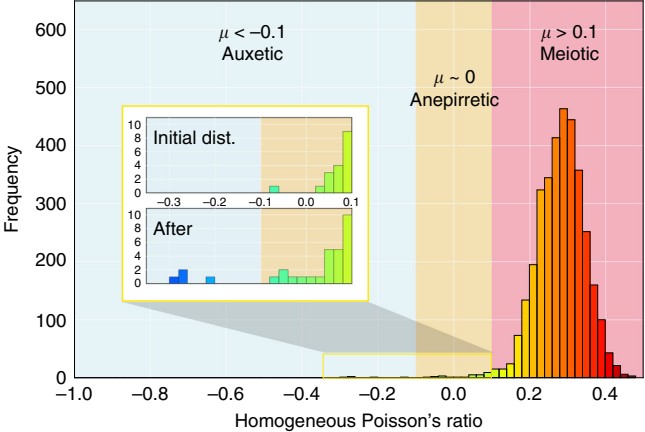

**Fig. 3** The distribution of homogeneous Poisson's ratio. Within the set of materials with elasticity data in the Materials Project database, meiotic materials far outnumber auxetic and anepirretic materials, which make up less than 1% of the total distribution. The inset shows the number of auxetic and anepirretic materials before and after the homologous structure search based on $\alpha$-cristobalite $SiO_2$

Project's high-throughput DFT elasticity workflow, which has been benchmarked to within 15% of available experimental measurements[25]. Of the 38 elastic tensor calculations, 30 successfully converged, passing all of the built-in filters of the workflow. These filters include removing materials that exhibit negative eigenvalues of the elastic tensor, which indicates soft modes and mechanical/dynamic instabilities at low temperatures. The average, homogeneous Poisson's ratio of the set was 0.07 and an average negative Poisson's ratio was obtained for 8 out of the 30 (Supplementary Table 1). In addition to $\alpha$-cristobalite, four polymorphs of $SiO_2$, two ternary metal vanadates, and a high-temperature phase of $AlPO_4$ were predicted to exhibit negative homogeneous Poisson's ratios. Another 5 materials of the 30 exhibited positive but near-zero homogeneous Poisson's ratios. (Table 1).

Instead of classifying the materials solely by the sign of the Poisson's ratio, we introduce a new criterion based on both sign and magnitude of $\mu$. Following the precedent set by Evans in assigning materials by Poisson's ratio under Greek terms[1], we have grouped structures with $|\mu| \leq 0.1$ separately from homogeneously auxetic materials. We denote this class of

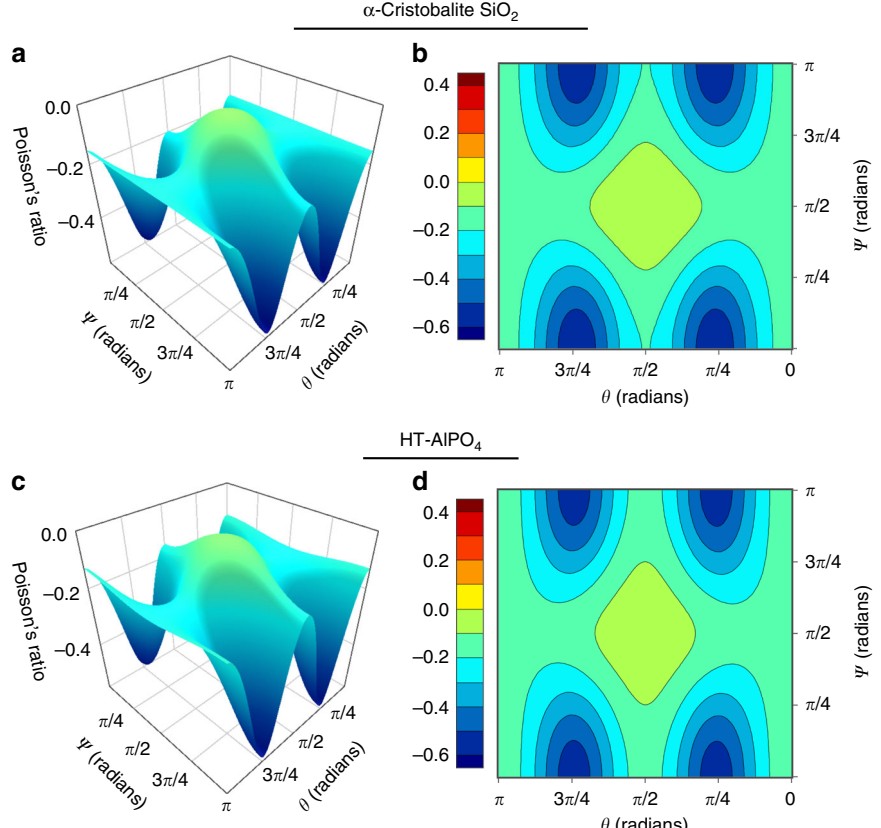

**Fig. 4** Directional Poisson's ratio maps for $\alpha$-cristobalite $SiO_2$ and HT-AlPO$_4$. $\nu_{32}$ was calculated from the elastic tensors of for $\alpha$-cristobalite $SiO_2$ **a**, **b** and HT-AlPO$_4$ **c**, **d** under basis rotations around the Eulerian $z$, $x'$, and $z''$ axes. These rotations were: $\phi = \pi/4$, $\theta = [0, \pi]$, and $\psi = [0, \pi]$ for $\alpha$-cristobalite and $\phi = 0$, $\theta = [0, \pi]$, and $\psi = [0, \pi]$ for HT-AlPO$_4$

materials 'anepirretic' after the Greek word $\alpha\nu\epsilon\pi\eta\rho\epsilon'\alpha\sigma\tau\sigma\varsigma$, which means "unaffected". These materials will maintain a nearly constant cross-section under linear-elastic tensile strain. The remaining, bulk majority of materials with $\mu > 0.1$ are denoted 'meiotic' after the corresponding antonym, $\mu\epsilon\iota\omega\tau\iota\kappa\acute{o}\varsigma$. Figure 3 shows the distribution of computed homogeneous Poisson's ratios in the Materials Project, which comprises the largest elastic tensor resource to date[25]. While the overwhelming majority (99%) of materials display meiotic behavior, the current search identified a statistically significant number of calculated negative or near-zero homogeneous Poisson's ratios. Of the 30 materials that passed Tier I and II, 1 known auxetic, 3 new auxetic, 9 new anepirretic, and 17 meiotic Poisson's ratios were predicted. This has greatly increased the total number of predicted auxetic and anepirretic materials in the MP database, as shown in the inset of Fig. 3. In the following section we discuss the resulting materials with negative Poisson's ratios (auxetic and anepirretic) in more detail.

**Resulting auxetic and anepirretic materials.** The results of the elastic analysis for each of the auxetic and anepirretic materials identified by the screening process are included in Table 1. Also included in Table 1, due to its pertinence to the role of chemistry in controlling the Poisson's ratio of materials, is GeO$_2$, a meiotic compound exhibiting a structure identical to $\alpha$-cristobalite.

Two novel silica polymorphs, denoted $SiO_2$(a) and (b), were identified as auxetic by the screening process. We note that $SiO_2$(a) is the enantiomer of $\alpha$-cristobalite $SiO_2$, hence it is not surprising that its elastic properties are close to those of its chiral partner. Coh and Vanderbilt investigated the structural stability of cristobalite and described how the $\alpha$ and $\beta$ structures may be

considered higher-symmetry special cases of a three-dimensional manifold with general $P2_12_12_1$ symmetry accessed by RUMs[19]. Both $SiO_2$(a) and (b) appear on this manifold along with $\alpha$ and $\beta$ cristobalite. Intuitively, the homogeneous Poisson's ratios of these structures fall between those of $\alpha$ and $\beta$-cristobalite $SiO_2$.

Two other silica polymorphs, denoted $SiO_2$(c) and (d), as well as $\beta$-cristobalite, denoted $SiO_2(\beta C)$, were calculated as anepirretic. Furthermore, two ternary metal vanadates; $FeV_3O_8$ and $CoV_3O_8$ were identified as anepirretic with predicted Poisson's ratios of $-0.05$ and $-0.04$, respectively. The four silica polymorphs $SiO_2$ (a)–(d) and the two ternary vandates are part of the growing number of materials in the Materials Project which are 'hypothetical' new compounds[31, 35], e.g., they have been generated using various transformation schemes, such as substitutions on existing materials, to explore novel chemistries and structures for different applications. Hence, there is no existing literature on synthesis and testing of these materials; however, they exhibit favorable formation energies and may be amenable for synthesis[34].

The potentially most interesting of the novel identified auxetic materials is a high-temperature phase of aluminum orthophosphate: HT-AlPO$_4$. This known polymorph exhibits a structure very similar to $\alpha$-cristobalite but has a lower crystal symmetry (C222$_1$) due to its stoichiometry. Our calculations predict it has a Poisson's ratio of $-0.28$, rivaling that of $\alpha$-cristobalite $SiO_2$. Berlinite AlPO$_4$ spontaneously transforms to HT-AlPO$_4$ at elevated temperatures, which remains metastable at room temperature[18, 36]. Although NAT-type aluminophosphate zeolites have been investigated for negative Poisson's ratio[37, 38], to our knowledge HT-AlPO$_4$ has not been experimentally investigated for auxetic behavior[39].

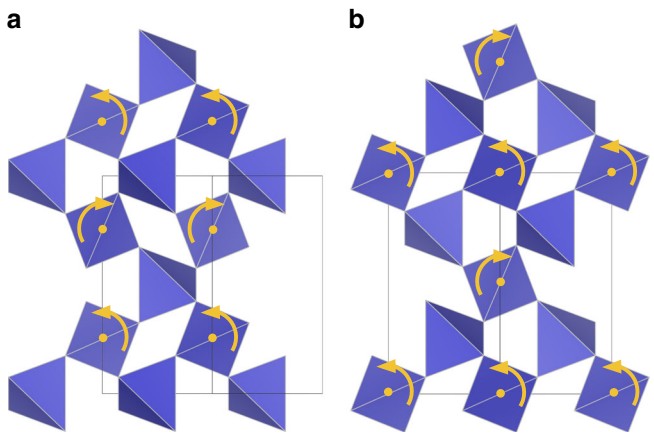

**Fig. 5** Tetrahedral rotations in $\alpha$-cristobalite $SiO_2$. Illustration of the rotational axes for the $SiO_4$ tetrahedra under tensile strain along the [001] direction as viewed from the **a** [110] direction and the **b** [$\bar{1}$10] direction in $\alpha$-cristobalite $SiO_2$

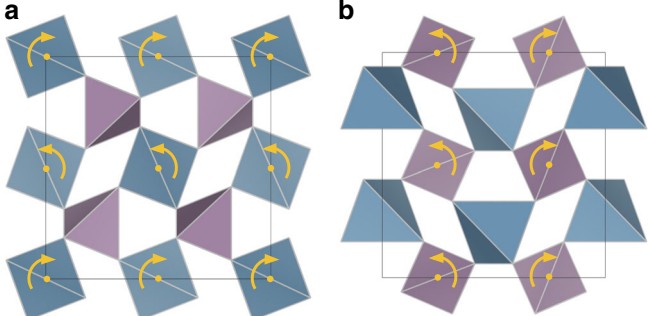

**Fig. 6** Tetrahedral rotations in HT-AlPO$_4$. Illustration of rotational axes of $AlO_4$ and $PO_4$ tetrahedra under tensile strain in the [001] direction. In **a** the rotational axis of the $AlO_4$ tetrahedra is depicted as viewed from the [100] direction whereas **b** shows the rotational axis of the $PO_4$ tetrahedra as viewed from the [010] direction

In order to study the extremal values of the anisotropic directional Poisson's ratio, $\nu_{ij}$, for the materials identified in Tier 3, we calculate $\nu_{32}$ over the entire range of possible orientations and compare to the known $\alpha$-cristobalite $SiO_2$. The elastic tensors were rotated around the Eulerian $z$, $x'$, and $z''$ directions between 0 and $2\pi$ radians at a resolution of $\pi/30$ and for each orientation, $\nu_{32}$ was recorded as a function of the rotation angles. Figure 4 shows the variation in the anisotropic directional Poisson's ratio $\nu_{32}$ for $\alpha$-cristobalite and HT-AlPO$_4$ over the subspace of orientations spanned by rotations about the $x'$ and $z''$ axes. In order to more directly compare the Poisson response of the two materials, the elastic tensor of $\alpha$-cristobalite was first rotated by $\pi/4$ around $z$ to align its orientation with that of HT-AlPO$_4$'s elastic tensor with respect to its conventional standard unit cell.

Indeed, $\nu_{max}$ is predicted as negative for both $\alpha$-cristobalite and HT-AlPO$_4$, implying that, similar to $\alpha$-cristobalite $SiO_2$, HT-AlPO$_4$ will behave auxetically when strained in any direction. We note that these results are consistent with other DFT calculations of the elastic properties of $\alpha$-cristobalite, which slightly overestimate the auxetic response such that $\nu_{max} < 0$, whereas Brillioun spectroscopy indicates $\nu_{max} = +0.08$[40]. Nevertheless, $\alpha$-cristobalite $SiO_2$ still exhibits overall auxetic behavior in experiment[12] and the same is expected for HT-AlPO$_4$. We find that $\nu_{min}$ appears on the $x'$–$z''$ rotation surfaces of $\alpha$-cristobalite and HT-AlPO$_4$ at coinciding rotations, hence the character of their Poisson behavior is highly similar. Figures 5 and 6 illustrate the tetrahedral rotations of the two materials for [001] tensile strains, which are also congruent in character. These predictions for RUM rotations under strain are consistent with the results of previous investigations into the origins of auxetic behavior in $\alpha$-cristobalite[14, 41]. Moreover, these soft rotational modes are also activated in the $\alpha$–$\beta$ transformation of cristobalite $SiO_2$[18]. $SiO_2$ (a) also exhibits a negative $\nu_{max}$, and hence global auxetic behavior, and $SiO_2$ (b) manifests a $\nu_{max}$ very close to 0.0, indicating auxetic behavior in a majority of directions.

Auxetic materials have conventionally been classified as "partially auxetic" if they possessed one or more negative directional Poisson's ratios or "completely auxetic" if they exhibited no positive values for $\nu_{ij}$[22]. While completely auxetic materials will always be homogeneously auxetic, partially auxetic materials can average to auxetic, anepirretic, or meiotic depending on the distribution of $\nu_{ij}$. Anepirretic materials typically exhibit directional Poisson's ratios that are distributed across both positive and negative values resulting in a homogeneous Poisson's ratio close to zero. As indicated in Table 1, a number of the final candidates display positive but near-zero homogeneous Poisson's ratios. The $SiO_2$ (c) polymorph as well as the ternary vandates display weakly negative Poisson's ratios while positive $\nu_{max}$. Uniformly small directional Poisson's ratios as well as combinations of positive and negative directions can minimize the magnitude of the homogeneous Poisson's ratio in anepirretic materials, which may be a useful materials design feature for applications where transverse strain is undesirable.

## Discussion

Accelerated discovery of novel materials with target properties is desirable to meet current and future material needs. In this study, we show how a tiered search, targeting specific structural motifs, can greatly accelerate the discovery of materials with unusual elastic behavior.

To date, very few inorganic crystalline materials other than $\alpha$-cristobalite $SiO_2$ are known to exhibit homogeneous auxetic behavior. Prior to this study, through the Materials Project calculations of elastic tensors, a tungsten oxide ($WO_3$) with weakly negative Poisson's ratio, $\mu = -0.08$, was serendipitously found among the first 2000 materials calculated[25]. This result represents an ~0.02% chance to discover a material with a negative homogeneous Poisson's ratio. Using the target screening described here, we find that almost one in five of the materials identified by the homologous structure search exhibit negative homogeneous Poisson's ratios and 3/30 are proposed new homogeneous auxetics. Hence, the approach delivers an improvement of three orders of magnitude.

It is important to note that the Materials Project database does not discern between materials that have been synthesized and those that remain hypothetical. Previously synthesized or known materials, identified by the tiered search as either auxetic or anepirretic, are HT-AlPO$_4$, $\beta$-cristobalite $SiO_2$, and GaPO$_4$. Both phosphates are metastable at room temperature and can be flux-grown or synthesized through annealing of quartz-structured AlPO$_4$ and GaPO$_4$ above the cristobalite transition temperature with subsequent quenching[36, 42]. However, $\beta$-cristobalite $SiO_2$ spontaneously transforms to $\alpha$-cristobalite upon cooling and must be chemically stabilized to exist at room temperature[43]. The remaining seven materials identified by the search as auxetic or anepirretic are hypothetical compounds with low formation energies which indicate a fair chance of successful synthesis[34]. The polymorphs $SiO_2$ (a), (c), and (d) are frameworks generated by simulated annealing[35] while $SiO_2$ (b) represents an early suggestion for the $\alpha$-cristobalite structure proposed in 1932. The four vanadates ($FeV_3O_8$, $CoV_3O_8$, $MnV_3O_8$, and $BVO_4$) originate from a variety of structure manipulations of known compounds,

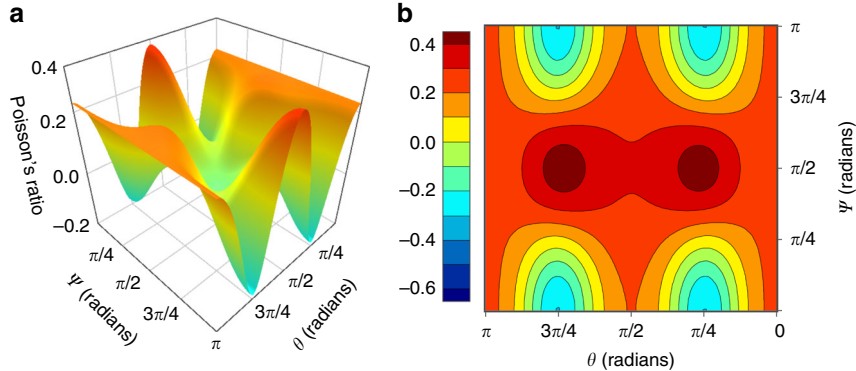

**Fig. 7** Directional Poisson's ratio map for $\alpha$-cristobalite GeO$_2$. The Poisson's ratio surface **a** and contour plot **b** for $\alpha$-cristobalite GeO$_2$ show the variation of $\nu_{32}$ as derived from the calculated elastic tensor under basis rotations around the Eulerian $z$, $x'$, and $z''$ axes. These rotations were: $\phi = \pi/4$, $\theta = [0, \pi]$, and $\psi = [0, \pi]$

such as orderings of disordered structures, and cation substitution in such frameworks[44]. We stress that the fact that these materials are hypothetical does not necessarily preclude the possibility that they can be made, and identifying them as auxetic may inspire design and discovery of similar compounds. Additionally, we note that the restriction on energy above the convex hull in Tier I of the screening process does not reflect an absolute limit for stability. As such, this screening method may exclude materials that can be synthesized and exploring higher energies may lead to the discovery of additional homogeneously auxetic materials of interest.

The presented work is built on the hypothesis that the structural motif—encompassed by rigid, corner-connected tetrahedra and open structures—is the dominating driver for auxetic or anepirretic behavior. As such, our search was intentionally agnostic to chemistry. While this hypothesis is clearly validated by the high success rate of the Tier III materials, chemistry also clearly plays a role in differentiating the Poisson's ratios of materials with similar structures. As an example, we offer $\alpha$-cristobalite GeO$_2$ which exhibits the exact same crystal structure, symmetry, and stoichiometry as $\alpha$-cristobalite SiO$_2$. Indeed, GeO$_2$ has previously been proposed as an auxetic material[9]. However, while auxetic directions are found in this material (see Fig. 7), the majority of strain directions result in positive Poisson's ratios and hence, its homogeneous Poisson's ratio is calculated as 0.10. Thus, we find that the compound most similar to alpha-cristobalite both in structure and chemistry is meiotic. This example is similarly mirrored by HT-AlPO$_4$ being auxetic while the highly related compound GaPO$_4$ is not. We speculate that small differences in bonding and bond lengths may be enough to influence the soft rotational modes that underlie auxetic behavior. Hence, this particular structural motif is found to be a highly correlated, but insufficient, descriptor for auxetic behavior. We speculate that additional or broader structural motifs could be employed as markers for potential auxetic and anepirretic behavior.

In summary, using a structure-matching procedure, together with stability and computational cost screening, drastically reduced the number of DFT calculations required to predict three previously unknown homogeneously auxetic compounds, whereof one is a known material; the high-temperature AlPO$_4$. Furthermore, our screening identified nine anepirretic materials, which are characterized by possessing a combination of both auxetic and meiotic directions resulting in an almost zero Poisson's ratio. The success of our approach suggests that negative or near-zero Poisson's ratio is a common feature of materials with cristobalite-like structures, such that a dramatically higher success rate is achieved if the structure motif is included in a

materials screening. However, we also note that chemistry is not unimportant as several $\alpha$-cristobalite-structured materials still exhibit meiotic behavior. We hope that the results of this study demonstrate the advantage of combining materials databases and intelligent materials design to accelerate materials discovery.

## Methods

**Ab initio calculations**. The high-throughput methodology and workflow used to calculate the elastic tensors was documented and benchmarked for all experimentally known elastic tensors as described previously in detail by de Jong et al.[25] In short, the elastic workflow begins with a structure optimization and subsequently calculates the stress of independent deformations of the optimized structure. Independent linear and shear strains between −1 and 1% are applied to the relaxed structures and the resultant stress tensors are calculated for each of these strain states. The calculations use first-principles DFT as implemented in the VASP code[45, 46]. Because the calculations require converged determinations of the bulk stress, a dense mesh of 7000 k-points per reciprocal atom and a cutoff energy of 700 eV for the plane-wave basis set are used. Exchange-correlation effects of the electronic density are accounted for using the Perdew–Burke–Ernzerhof functional[47]. In addition, standard Materials Project Hubbard U corrections are used for a number of transition metal oxides[48], as documented and implemented in the pymatgen VASP input sets[32]. Resulting elastic tensors are filtered on the basis of multiple criteria for mechanical stability and reliability as previously documented[25]. The full content of the workflow and analytical tools used to generate this and all other MP elastic tensor data may be found in the MPWorks and pymatgen code packages at github.com/materialsproject.

**Calculation of Poisson's ratios**. The homogeneous approximation of the Poisson's ratio was calculated from the elements of the calculated elastic tensors as:

$$\mu = \frac{3K_{VRH} - 2G_{VRH}}{6K_{VRH} + 2G_{VRH}} \qquad (1)$$

where $K_{VRH}$ and $G_{VRH}$ are the Voigt–Reuss–Hill averaged bulk and shear moduli, respectively[25]. Furthermore, the anisotropic Poisson response of materials were investigated through the calculation of the directional Poisson ratios, $\nu_{ij}$, determined from the coefficients of the Voigt compliance tensor (inverse of the elastic tensor), $S$, as:

$$\nu_{ij} = -\frac{S_{ij}}{S_{ii}} \qquad (2)$$

where the value for $\nu_{ij}$ represents the ratio of deformation in the principal direction $j$ to a uniaxial strain applied along the orthogonal principal direction $i$ in the basis frame of the elastic tensor[26]. The global minimum and maximum directional Poisson's ratios were found by rotating the original elastic tensor through the range of possible orientations via the Euler–Rodriguez and Cayley method and Euler angle rotations around the $z$, $x'$, and $z''$ axes at a resolution of $\pi/30$ radians[49].

**Determination of tetrahedral rotations**. The tetrahedral rotations displayed in Figs. 5 and 6 were derived from ideal tensile strain calculations[50], in which the equilibrium bulk structures are deformed in a single direction and optimized isotropically in the perpendicular directions. This allowed for an estimate of the structural response to a uniaxial load, and further reinforces the prediction of auxetic behavior in elastic tensor results.

**Structure comparisons**. The structure-comparison algorithm described in this study was implemented using the structure matcher in Pymatgen[32]. Structure-matching proceeds by: 1. reducing to unit cells and comparing the total number of sites, 2. further reducing to Niggli cells and scaling by volume, 3. removing oxidation states, and 4. comparing, within specified tolerances, the lattice angles and atomic positions of all permutations of valid lattices. A full description of the algorithm is provided. (Supplementary Methods) The structure matcher was initialized with a framework comparator, which ignores species and oxidation state when comparing sites between two structures, and the following structure-matching parameters: fractional length tolerance = 0.2, fraction of average free length per atom = 0.5, and angle tolerance = 5°.

**Data availability**. The code and data used to develop the screening process in this study are available at http://pymatgen.org and https://materialsproject.org. Any other information supporting the conclusions presented is available from the authors upon request.

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

## Acknowledgements

This work was intellectually led by the Department of Energy (DOE) Basic Energy Sciences (BES) program—the Materials Project—under Grant No. EDCBEE. This research used resources of the National Energy Research Scientific Computing Center, which is supported by the Office of Science of the U.S. Department of Energy under Contract No. DEAC02-05CH11231. We thank Ioannis Petousis for suggesting the terms "anepirretic" and "meiotic".

## Author contributions

J.D. developed the screening process and wrote its code, performed elastic constant calculations, and worked on data analysis. J.M. performed elastic constant calculations

and was involved in the data analysis. M.D.J was involved in the planning of the screening process. K.P. was involved in supervising and planning the work. J.D., J.M, and K.P. all contributed to the writing of the manuscript.

## Additional information

**Competing interests:** The authors declare no competing financial interests.

