## [Peer Review File · Nature Communications]

Reviewers' comments:

Reviewer #1 (Remarks to the Author):

Generally this is an excellent paper with some very interesting hypothetical conclusions (which will need to be tested experimentally at some point by someone). It is to be hoped that this predictive work may stimulate new experimental chemistry to make novel auxetic materials. It is generally well written and well organised. I fully support its publication and the following comments are intended to suggest some improvements in clarity.

There is some confusion in the Abstract and opening paragraph about the definition of "auxetic":

Abstract

"Most known auxetic materials are either porous foams or artificial macrostructures" This is a highly inaccurate statement. Firstly, many microstructures and composites demonstrate at least partial "auxeticity". More importantly, previous work in the literature (several papers quoted in this paper) have shown that many common crystalline materials are at least partially auxetic, in specific directions.

Pg 1 para 1 Use ν (nu) for Poisson's ratio throughout, or make it clear from the beginning that μ (mu) is being used for the homogeneously averaged Poisson's ratio.

Pg1 para2 "Auxetic defines a class of....that exhibit an average negative Poisson's ratio" This is also wrong. Auxetic does not mean an average negative Poisson's ratio it means a material that demonstrates any negative Poisson's ratio behaviour – with earlier papers referring to partial or total "auxeticity" and with various specified limits and ranges.

In fact the authors go on to say this later in the paragraph but do not make this clear at the start.

The predictions of Poisson's ratio rely on the accuracy of the elastic tensors in the database – many of which are not all experimentally determined and are notoriously sensitive to inaccuracies. This needs to be stated much more clearly.

It is odd that the authors to choose to specify what appears to be a fairly arbitrary limit on formation energy (to eliminate materials that may be "challenging to synthesize") since it is quite possible that the energetics required for "fully auxetic" materials may actually require the material to be "difficult to synthesize". Something around this is probably worth saying in the discussion.

The screening and optimising methods would benefit from being described in more detail (in supplementary?) It would also be useful if they specify the extrema of the Poisson's ratios and in what directions they may be found (in supplementary?).

Did the screening process include any equivalent structures for plutonium (a known partially auxetic material)?

Page 4 para 1 "Amendable" to be corrected to "amenable"?

Pg 6 III Methods section

Para1 sentence 1 "The homologous.....etc" – incomprehensible!

Reviewer #2 (Remarks to the Author):

In the article the authors were able to identify new auxetic materials (partially auxetics and completely auxetics) and investigate variability of Poisson's ratio for them. The angular dependence of Poisson's ratio for some materials have been plotted. Following remarks can be noted:

1. In the introduction of article study of variability of Poisson's ratio weakly reflected for alpha-cristobalite. The authors have been given only reference to [8]. J.N. Grima, K.E. Evans and A. Alderson have a large series of works devoted to the elastic properties of alpha-cristobalite:

- J.N. Grima, V. Zammit, R. Gatt, D. Attard, C. Caruana, G. Trevor. On the role of rotating tetrahedra for generating auxetic behavior in NAT and related systems. *Journal of Non-Crystalline Solids*. 2008. 354(35-39), P.4214-4220. DOI: 10.1016/j.jnoncrysol.2008.06.081
 - J.N. Grima, R. Gatt, A. Alderson and K.E. Evans, An alternative explanation for the negative Poisson's ratios in alpha-cristobalite, *Mater. Sci & Eng. A*, 423 (2006) p. 219-224.
 - J.N. Grima, R. Gatt, A. Alderson and K.E. Evans, On the origin of auxetic behaviour in the silicate alpha-cristobalite, *J. Mater. Chem.*, 15 (2005) p. 4003 - 4005.
 - J.N. Grima, V. Zammit, R. Gatt, D. Attard, C. Caruana and T.G. Chircop Bray, On the role of rotating tetrahedra for generating auxetic behaviour in NAT and related systems, *J. Non-Cryst. Sol.*, 354 (2008) p. 4214-4220.
 - J.N. Grima, J.J. Williams, R. Gatt, V. Zammit, A. Alderson, R.I. Walton and K.E. Evans, Natrolite: A zeolite with negative Poisson's ratios, *J. Appl. Phys.*, 101 (2007) 086102.
 - R. Gatt, L. Mizzi, K.M. Azzopardi, J.N. Grima. A force-field based analysis of the deformation mechanism in α -cristobalite. *Phys. Status Solidi B*. 2015. 252(7) P.1479-1485 DOI: 10.1002/pssb.201552133
 - K. M. Azzopardi, J. P. Brincat, J. N. Grima, R. Gatt Anomalous elastic properties in stishovite. *RSC Adv.*, 2015,5, P.8974-8980 DOI: 10.1039/C4RA12072H
 - F. Nazaré, A. Alderson. Models for the prediction of Poisson's ratio in the ' α -cristobalite' tetrahedral framework. *Phys. Status Solidi B*. 2015. 252(7), P.1465-1478 DOI: 10.1002/pssb.201451732
- Review - H.M.A. Kolken and A.A. Zadpoor. Auxetic mechanical metamaterials. *RSC Adv*. 2017. 7(9), P. 5111-5129. DOI: 10.1039/C6RA27333E
- Discussion of these articles and their comparison with the results of this article is necessary.

2. In the article authors investigate Poisson's ratio of materials. For this purpose they use the theory of elasticity of an anisotropic body. At the same time in introduction crystal auxetics are not discussed. The only reference is given (Ref.[15]). However, many researchers have studied the auxetic properties of crystals. At the moment more than 400 crystal auxetics are revealed among 2000 crystals experimental data for which are listed in Landolt-Bornstein Handbook. These crystal auxetics are given in:

- R.H. Baughman, J.M. Shacklette, A.A. Zakhidov, S. Stafström. Negative Poisson's ratios as a common feature of cubic metals. *Nature*. 1998. 392(6674). P. 362-365. DOI: 10.1038/32842
- R.V. Goldstein, V.A. Gorodtsov, D.S. Lisovenko. Auxetic mechanics of crystalline materials. *Mech. Solids*. 2010. 45(4). P.529-545. DOI: 10.3103/S0025654410040047

- R.V. Goldstein, V.A. Gorodtsov, D.S. Lisovenko. Classification of cubic auxetics. Phys. Status Solidi B. 2013. 250(10), P.2038-2043. DOI: 10.1002/pssb.201384233
- R.V. Goldstein, V.A. Gorodtsov, D.S. Lisovenko, M.A. Volkov. Negative Poisson's ratio for cubic crystals and nano/microtubes. Phys. Mesomech. 2014. 17(2). P.97-115. DOI: 10.1134/S1029959914020027
- R.V. Goldstein, V.A. Gorodtsov, D.S. Lisovenko. Young`s modulus and Poisson`s ratio for seven-constant tetragonal crystals and nano/microtubes. Phys. Mesomech. 2015. 18(3). P.213-222. DOI: 10.1134/S1029959915030054
- R.V. Goldstein, V.A. Gorodtsov, D.S. Lisovenko, M.A. Volkov. Auxetics among 6-constant tetragonal crystals. Letters on Materials. 2015. 5(4). P.409-413. DOI: 10.22226/2410-3535-2015-4-409-413
- R.V. Goldstein, V.A. Gorodtsov, D.S. Lisovenko. The elastic properties of hexagonal auxetics under pressure. Phys. Status Solidi B. 2016. 253(7). P.1261-1269. DOI: 10.1002/pssb.201600054
- Epishin A.I., Lisovenko D.S. Extreme values of Poisson's ratio of cubic crystals. Technical Physics. 2016. 61(10). P.1516-1524. DOI: 10.1134/S1063784216100121
- R.V. Goldstein, V.A. Gorodtsov, D.S. Lisovenko, M.A. Volkov. Mechanical characteristics for seven-constant rhombohedral crystals and their nano/microtubes. Letters on Materials. 2016. 6(2). P.93-97. DOI: 10.22226/2410-3535-2016-2-93-97

Discussion of these articles is necessary.

3. The authors propose a new classification of crystal auxetics (Auxetic-Anepirretic-Meiotic). Other classifications already exist for crystal auxetics:

- a) for polycrystals: auxetic $-1 < \nu < 0$ and nonauxetic $0 < \nu < 0.5$;
- b) for monocrystals: nonauxetic ($\nu_{min} > 0, \nu_{max} > 0$), partially auxetic ($\nu_{min} < 0, \nu_{max} > 0$), completely auxetic ($\nu_{min} < 0, \nu_{max} < 0$).

They are published in

- T.C.T. Ting, D.M. Barnett. Negative Poisson's ratios in anisotropic linear elastic media. J. Appl. Mech. 2005. 72(6), P. 929-931. DOI: 10.1115/1.2042483
- A.C. Branka, D.M. Heyes, K.W. Wojciechowski. Auxeticity of cubic materials. Phys. Status Solidi B. 2009. 246(9), P. 2063-2071. DOI: 10.1002/pssb.200982037
- A.C. Branka, D.M. Heyes, K.W. Wojciechowski. Auxeticity of cubic materials under pressure. Phys. Status Solidi B. 2011. 248(1). P. 96-104. DOI: 10.1002/pssb.201083981
- Goldstein R.V., Gorodtsov V.A., Lisovenko D.S. Classification of cubic auxetics. Phys. Status Solidi B. 2013. 250(10), P.2038-2043. DOI: 10.1002/pssb.201384233

Discussion of these classifications should be added to article.

4. The authors have shown that eight crystals can have a negative average Poisson's ratio. Negative average Poisson's ratio was obtained for some of the materials in [Goldstein R.V., Gorodtsov V.A., Lisovenko D.S. Average Poisson's ratio for crystals. Hexagonal auxetics. Letters on Materials. 2013. 3(1). P.7-11.]. Discussion of this article should be added in the introduction of article.

5. The results of Table 1 show that the GeO₂ crystal has a negative Poisson's ratio. Early auxeticity for this crystal was found in [Goldstein R.V., Gorodtsov V.A., Lisovenko D.S.,

Volkov M.A. Auxetics among 6-constant tetragonal crystals. Letters on Materials. 2015. 5(4). P.409-413]. Page 1, right column «as cubic metals^{15} and layered materials.^{2,13,16}». In [Goldstein R.V., Gorodtsov V.A., Lisovenko D.S. The elastic properties of hexagonal auxetics under pressure. Phys. Status Solidi B. 2016. 253(7). P.1261-1269] it is shown that MoS₂ is layered auxetic. Comparison of the results should be added to article.

Due to above mentioned, this article requires revision for its publication in the journal «Nature Communications».

Reviewer #3 (Remarks to the Author):

The authors report on the computational prediction of new auxetic materials. They have been used high-throughput computational methods to accelerate the discovery of materials with negative Poisson's ratio (known as auxetic materials). They proposed a strategy to aggressively reduce the data set from 67,000 materials to 38 candidates according the structure match compared with the alpha-cristobalite. This successfully lead to the prediction of three previously unidentified auxetic materials and a number of materials with near-zero Poisson's ratio.

The presented results are very encouraging. For a long time, alpha-cristobalite is the only known crystalline materials possessing auxetic behavior. Materials with exotic properties were discovered by serendipity from time to time. Nowadays, with the help of database and proper machine learning strategy, the authors has shown that the chance of finding the materials with optimum target properties is much higher with affordable computational cost. This will find a broad interest from various area of the materials science community.

Thus, I am supportive in publishing the manuscript in NComm. However, more work is needed to bring the contributions to a level that can be appreciated by the broad audience.

1) It is not surprising to identify the HT-AlPO₄ as the auxetic materials with competitive Poisson's ratio. AlPO₄ is exactly isoelectronic to SiO₂. And they have the same packing. Such analogue should be expected by any materials researcher even without calculation. Before the era of high-throughput in materials sciences, such work based on chemically similarity has been the dominant tool for human-learning materials discovery. The reason why AlPO₄ was neglected is perhaps because crystalline auxetic materials are not as hot as li-battery, thermoelectric materials. Following AlPO₄, materials like BPO₄, BAsO₄ also have the similar structures. I wonder if any of them fall to the Tier II in the prediction. The results on Co(Fe/Mn)V₃O₈ seems to me more valuable (since no one would consider such materials immediately as the analogue to cristobalite). There should be some words to discuss how the approach is different from the traditional approach (more efficient or robust?). In addition, the authors should give the information of tier II materials in the SI. I didn't find it in my review.

2) In Tier II, the author used a strategy to narrow down the number of candidate structures

from 36,000 to 38 based on similarity match to alpha-cristobalite. I wonder how strict the criteria are. It looks like rather an aggressive reduction. In the database, there should exist plenty of structures featured by packing of corner shared tetrahedra. I think that's why the uncovered SiO₂(a-c) are structurally similar to alpha/beta-cristobalite. The authors should present the way to define the structure matchers in more details, and evaluate the sensitivity due to the choice of different numerical settings. Furthermore, it occurs to me if the criterion might be too strict. Indeed, it is widely believed that the auxetic behavior is largely due to the rigidity of SiO₄ tetrahedra and their unusual packing in 3D. But if one looks for the category of 'auxetic materials', I guess this criterion of 'corner sharing tetrahedral' might be too strict. The authors should mention this as well. In principle, it would be more interesting to uncover a new motif different to alpha-cristobalite SiO₂.

3) It is good that the authors mention the synthesizability of these materials in the end. For the general audience, it would be good to mention the possible applications of crystalline auxetic materials as well.

4) The citation is not well balanced. The authors cited too much work related to 'Materials Projects'. There should be some spaces for other work similar to materials projects.

We sincerely thank the reviewers for their time, valuable feedback, and the opportunity to improve our manuscript. The revised manuscript highlights the changes in red and we are submitting a Supplemental Information (SI) to provide further details as requested by the reviewers. Below we provide a point-by-point response to the reviewers comments:

Referee 1

We are very gratified that the reviewer found our manuscript well-written and well-organized. We have carefully considered each of the points suggested by the referee (see below) and made changes where appropriate.

1. There is some confusion in the abstract and opening paragraph about the definition of auxetic: Abstract: "*Most known auxetic materials are either porous foams or artificial macrostructures*" This is a highly inaccurate statement. Firstly, many microstructures and composites demonstrate at least partial auxeticity. More importantly, previous work in the literature (several papers quoted in this paper) have shown that many common crystalline materials are at least partially auxetic, in specific directions.
2. Use ν (nu) for Poissons ratio throughout, or make it clear from the beginning that μ (mu) is being used for the homogeneously averaged Poissons ratio.
3. "Auxetic defines a class of that exhibit an average negative Poissons ratio" This is also wrong. Auxetic does not mean an average negative Poissons ratio it means a material that demonstrates any negative Poissons ratio behaviour with earlier papers referring to partial or total auxeticity and with various specified limits and ranges. In fact the authors go on to say this later in the paragraph but do not make this clear at the start.

We agree that our use of terminology was somewhat confusing. To remedy this, we have revised the introduction and other relevant sections to point out the distinction between auxetic materials (which have one or more negative directional Poisson's ratios) and homogeneously auxetic materials (materials which exhibit a negative Poisson's ratio under the Voigt-Reuss-Hill averaging scheme). For consistency, we have replaced the term "Poisson's ratio" with the term "homogeneous Poisson's ratio" in many instances to improve clarity. Additionally, we explicitly define μ as the homogeneous Poisson's ratio early in the introduction.

4. The predictions of Poissons ratio rely on the accuracy of the elastic tensors in the database, many of which are not all experimentally determined and are notoriously sensitive to inaccuracies. This needs to be stated much more clearly.

We concur that more explicit commentary on the accuracy of DFT-based methods is appropriate, and have added text describing how previous benchmarks have reproduced experimental bulk and shear moduli.[1] In addition, we include a statement on the sensitivity of our calculations and a further comment on how this relates to their computational cost:

"Benchmarks of the Materials Project elastic tensor workflow suggest that it reproduces elastic-tensor derived bulk and shear moduli well; within 15% of experimentally measured values. However, the workflow requires computationally demanding input parameters to ensure this level of accuracy.[1] It is currently projected to take years to calculate the elastic tensor for all known crystalline compounds using available supercomputing resources. Hence, approaches which prioritize certain compounds based on structure-property descriptors are useful to accelerate the rate of discovery. For example, predictive statistical learning methods have been used to estimate bulk and shear moduli of materials in the MP database, illustrating that existing DFT-based data on elasticity and structure can be leveraged to guide the search for specific elastic behavior.[2]"

5. It is odd that the authors to choose to specify what appears to be a fairly arbitrary limit on formation energy (to eliminate materials that may be challenging to synthesize) since it is quite possible that the energetics required for fully auxetic materials may actually require the material to be difficult to synthesize. Something around this is probably worth saying in the discussion.

First of all, we apologize. There was an error in the text - which has now been corrected with respect to the cutoff energy limit. We applied a 100 meV/atom (not 30 meV/atom) limit to the thermodynamic stability in Tier I. While restrictive, the 100 meV/atom limit is not quite arbitrary. Recently, in [3], it is shown that approximately 80% of all known, naturally existing as well as laboratory-synthesized oxides are within 100 meV/atom of the ground state at zero Kelvin. The limit was hence chosen deliberately to prioritize materials that may be more easily synthesized. As pointed out by the referee, the opportunity to discover more auxetic materials by looking to higher energies is a possible path forward towards finding new auxetic materials. In particular, as was noted in [3], more covalent systems such as phosphides and nitrides allow for higher metastability limits, which will be the focus of future work. We have revised the discussion section to address this point:

"Additionally, we note that the restriction on energy above the convex hull in Tier I of the screening process does not reflect an absolute limit for stability. As such, this screening method may exclude materials that can be synthesized and exploring higher energies may lead to the discovery of additional homogeneously auxetic materials of interest."

6. The screening and optimizing methods would benefit from being described in more detail (in supplementary?) It would also be useful if they specify the extrema of the Poissons ratios and in what directions they may be found (in

supplementary?).

Agreed. We have included the structure matching algorithm used in the described screening process in the Supplementary Information.

7. Did the screening process include any equivalent structures for plutonium (a known partially auxetic material)?

Our screening process did not include any plutonium-containing compounds, due to ongoing debate and disagreement on whether standard-DFT is appropriate to describe these heavier, radioactive elements [4, 5] To clarify, a full list of the Tier II materials is included in the Supplemental Information.

8. Page 4 para 1 Amendable to be corrected to amenable?

Thank you for pointing this out. Amendable on page 4 paragraph 1 has been changed to amenable.

9. Pg 6 Methods section Para 1 sentence 1 The homologous..etc incomprehensible!

In an effort to describe our methods more clearly, we have changed the sentence to the following:

"The structure-comparison algorithm described in this study was implemented using the structure matcher in Pymatgen.[6] Structure matching proceeds by: 1. reducing to unit cells and comparing the total number of sites, 2. further reducing to Niggli cells and scaling by volume, 3. removing oxidation states, and 4. comparing, within specified tolerances, the lattice angles and atomic positions of all permutations of valid lattices. The structure matcher was initialized with a framework comparator, which ignores species and oxidation state when comparing sites between two structures, and the following structure matching parameters: fractional length tolerance = 0.2, fraction of average free length per atom = 0.5, and angle tolerance = 5°."

Additionally, we have included the structure matcher algorithm in its entirety in the Supplemental Information.

Referee 2

We appreciate the careful review and information provided by the reviewer. We have considered each of the points suggested (see below) and made appropriate changes.

1. In the introduction of article study of variability of Poissons ratio weakly reflected for alpha-cristobalite. The authors have been given blue reference to [8].

J.N. Grima, K.E. Evans and A. Alderson have a large series of works devoted to the elastic properties of alpha-cristobalite:

[Sources listed]

Discussion of these articles and their comparison with the results of this article is necessary.

Thank you for bringing these resources to our attention. We have revised the introduction and discussion sections to incorporate information contained in these articles. We have also included references to these works in our sections on the RUM model for auxetic behavior and the elastic properties of alpha-cristobalite and compared our predictions with these results in our discussion section as follows:

"Indeed, the prevailing theory for auxetic behavior in α -cristobalite and other inorganic materials relies on rigid unit modes (RUM's) in which connected polyhedra rotate without deforming. (see Fig 1.) [7, ?, 8, 9, 10, 11]"
and

"These predictions for RUM rotations under strain are consistent with the results of previous investigations into the origins of auxetic behavior in α -cristobalite. [7, 12]"

2. In the article authors investigate Poissons ratio of materials. For this purpose they use the theory of elasticity of an anisotropic body. At the same time in introduction crystal auxetics are not discussed. The only reference is given (Ref.[15]). However, many researchers have studied the auxetic properties of crystals. At the moment more than 400 crystal auxetics are revealed among 2000 crystals experimental data for which are listed in Landolt-Bornstein Handbook. These crystal auxetics are given in:

[Sources listed]

Discussion of these articles is necessary.

Thank you for this suggestion. We have added a discussion on crystalline auxetic materials and other work concerning these structures in the introduction. Introduction, Page 1 Paragraph 2:

"A number of crystalline materials have been found to exhibit negative Poisson's ratios in certain directions associated with specific features of their crystal structures.[13, 14, 15, 16]"

3. The authors propose a new classification of crystal auxetics (Auxetic-Anepirretic-Meiotic). Other classifications already exist for crystal auxetics:

- a) for polycrystals: auxetic $-1 < \nu < 0$ and nonauxetic $0 < \nu < 0.5$;
 b) for monocrystals: nonauxetic ($\nu_{min} > 0, \nu_{max} > 0$), partially auxetic ($\nu_{min} < 0, \nu_{max} > 0$), completely auxetic ($\nu_{min} < 0, \nu_{max} < 0$).

They are published in

[Sources listed]

Discussion of these classifications should be added to article.

This is an excellent point. We have added discussion of these classifications and how our naming system relates to them in our results section.

"Auxetic materials have conventionally been classified as "partially auxetic" if they possessed one or more negative directional Poisson's ratios or "completely auxetic" if they exhibited no positive values for ν_{ij} . [17] While completely auxetic materials will always be homogeneously auxetic, partially auxetic materials can be auxetic, anepirretic, or meiotic depending on the distribution of ν_{ij} . Anepirretic materials exhibit directional Poisson's ratios that are distributed across both positive and negative values resulting in a homogeneous Poisson's ratio close to zero."

4. The authors have shown that eight crystals can have a negative average Poissons ratio. Negative average Poissons ratio was obtained for some of the materials in [Goldstein R.V., Gorodtsov V.A., Lisovenko D.S. Average Poisson's ratio for crystals. Hexagonal auxetics. Letters on Materials. 2013. 3(1). P.7-11.]. Discussion of this article should be added in the introduction of article.

5. The results of Table 1 show that the GeO2 crystal has a negative Poisson's ratio. Early auxeticity for this crystal was found in [Goldstein R.V., Gorodtsov V.A., Lisovenko D.S., Volkov M.A. Auxetics among 6-constant tetragonal crystals. Letters on Materials. 2015. 5(4). P.409-413]. Page 1, right column as cubic metals¹⁵ and layered materials^{2,13,16}. In [Goldstein R.V., Gorodtsov V.A., Lisovenko D.S. The elastic properties of hexagonal auxetics under pressure. Phys. Status Solidi B. 2016. 253(7). P.1261-1269] it is shown that MoS2 is layered auxetic. Comparison of the results should be added to article.

Thank you for bringing these resources to our attention. We have included references to them in the introduction and results section.

"Goldstein et al. report negative average Poisson's ratios for a small number of tetragonal and cubic materials calculated from previously published elastic constants. [17, 18] These include C_{16} FeGe₂, elemental Ba, and solid solutions based on rocksalt-structured SmS. However, we note that other reference work cites the Poisson's ratio of Ba as 0.28[19] and the calculated Poisson's ratio of C_{16} FeGe₂ is obtained as 0.24[1].

"As an example, we offer α -cristobalite GeO₂ which exhibits the exact same crystal structure, symmetry, and stoichiometry as α -cristobalite SiO₂ and has

been previously proposed as an auxetic material.[16]”

Referee 3

Thank you for your supportive review and excellent advice.

1. It is not surprising to identify the HT-AlPO₄ as the auxetic materials with competitive Poissons ratio. AlPO₄ is exactly isoelectronic to SiO₂. And they have the same packing. Such analogue should be expected by any materials researcher even without calculation. Before the era of high-throughput in materials sciences, such work based on chemically similarity has been the dominant tool for human-learning materials discovery. The reason why AlPO₄ was neglected is perhaps because crystalline auxetic materials are not as hot as li-battery, thermoelectric materials. Following AlPO₄, materials like BPO₄, BAsO₄ also have the similar structures. I wonder if any of them fall to the Tier II in the prediction. The results on Co(Fe/Mn)V₃O₈ seems to me more valuable (since no one would consider such materials immediately as the analogue to cristobalite). There should be some words to discuss how the approach is different from the traditional approach (more efficient or robust?). In addition, the authors should give the information of tier II materials in the SI. I didnt find it in my review.

Indeed all valid points. We have debated among ourselves the influence of structure versus chemistry. As we note in the Discussion, clearly structural similarity and chemistry correlate with similar Poisson’s ratio; however, the examples of GeO₂ and GaPO₄ stand out as important outliers. Similar to AlPO₄, both are isoelectronic to α -cristobalite SiO₂, yet their homogeneous Poissons ratios are predicted to be positive. The evidence gathered in this study suggests that while structure is an important descriptor, it is not sufficient to predict auxeticity.

The point about inclusiveness in Tier II is well taken - as pointed out by Reviewer 1 as well. Indeed, both BPO₄ and BAsO₄ were identified by the similar structures search, however our calculations predict homogeneous Poisson’s ratios of +0.12 for BPO₄ and +0.16 for BaSO₄. A full list of the Tier II materials is included in the new Supplemental Information section.

2. In Tier II, the author used a strategy to narrow down the number of candidate structures from 36,000 to 38 based on similarity match to alpha-cristobalite. I wonder how strict the criterions are. It looks like rather an aggressive reduction. In the database, there should exist plenty of structures featured by packing of corner shared tetrahedra. I think thats why the uncovered SiO₂(a-c) are structurally similar to alpha/beta-cristobalite. The authors should present the way to define the structure matchers in more details, and evaluate the sensitivity due to the choice of different numerical settings. Furthermore, it occurs to me if the criterion might be too strict. Indeed, it is widely believed that the auxetic

behavior is largely due the rigidity of SiO₄ tetrahedra and their unusual packing in 3D. But if one looks for the category of anepirretic materials, I guess this criterion of corner sharing tetrahedral might be too strict. The authors should mention this as well. In principle, it would be more interesting to uncover a new motif different to alpha-cristobalite SiO₂.

These criteria are indeed quite strict (as noted by Reviewer 1 as well), especially in the reduction from Tier I to Tier II, which retained only one material out of a thousand. We deliberately narrowed the field of candidate materials for two reasons: i) the expensive nature of elastic tensor calculations and ii) the effort required to synthesize new compounds. Each tensor requires on average 3000 CPU-hrs and we have a throughput of 120 compounds per month. However, the reviewer's point is well made and we emphasize in the Discussion that additional compounds with negative Poisson's ratios may well be found outside the employed search criteria and remain viable candidates for new search methodologies in the near future. Furthermore, detailed information regarding the structure matchers algorithm can be found in the supplementary information.

"Hence, while this particular structural motif is found to be highly correlated with auxetic behavior, it does not always result in negative Poisson's ratio. We speculate that broader structural motifs could be employed as markers for potential auxetic and anepirretic behavior."

3. It is good that the authors the synthesizability of these materials in the end. For the general audience, it would be good to mention the possible applications of crystalline auxetic materials as well.

We would like to thank the referee for their positive comment and their advice on how we might make this paper more accessible to a broad audience. We have addressed the possible applications for auxetic materials in the introduction of our paper. (See Page 1, Paragraph 1)

These materials have been shown to possess enhanced hardness and toughness, as well as absorb vibrations and sound better than their non-auxetic counterparts.[11, 8] As a result, the atypical elastic behavior of auxetic materials is enabling advancements in a broad range of technologies such as impact-resistant composites, extremely precise sensors, tougher ceramics, and high-performance armor. [20, 8, 21, 11]

We hope that this is a sufficient description of the important applications for auxetic materials.

4. The citation is not well balanced. The authors cited too much work related to Materials Projects. There should be some spaces for other work similar to materials projects.

This is valuable constructive criticism. We have not used any other resource, but we have added a reference to OQMD and AFLOWlib, which both provide free access to structure information of materials.

References

- [1] de Jong, M. *et al.* Charting the complete elastic properties of inorganic crystalline compounds. *Scientific Data* **2** (2015).
- [2] de Jong, M. *et al.* A statistical learning framework for materials science: application to elastic moduli of k-nary inorganic polycrystalline compounds. *Scientific Reports* **6** (2016).
- [3] Sun, W. *et al.* The thermodynamic scale of inorganic crystalline metastability. *Sci. Adv.* **2**, e1600225 (2016).
- [4] Söderlind, P., Zhou, F., Landa, A. & Klepeis, J. E. Phonon and magnetic structure in δ -plutonium from density-functional theory. *Nature Publishing Group* 1–6 (2015).
- [5] Lashley, J. C., Lawson, A., McQueeney, R. J. & Lander, G. H. Absence of magnetic moments in plutonium. *PHYSICAL REVIEW B* 1–12 (2005).
- [6] Ong, S. P. *et al.* Python Materials Genomics (pymatgen): A robust, open-source python library for materials analysis. *Computational Materials Science* **68**, 314–319 (2013).
- [7] Grima, J. N., Gatt, R., Alderson, A. & Evans, K. E. An alternative explanation for the negative Poisson’s ratios in alpha-cristobalite. *Materials Science and Engineering A* **423**, 219–224 (2006).
- [8] Evans, K. E. & Alderson, A. Auxetic materials: Functional materials and structures from lateral thinking! *Advanced Materials* **12**, 617–628 (2000).
- [9] Williams, J. J., Smith, C. W., Evans, K. E., Lethbridge, Z. A. & Walton, R. I. An analytical model for producing negative Poisson’s ratios and its application in explaining off-axis elastic properties of the NAT-type zeolites. *Acta Materialia* **55**, 5697–5707 (2007).
- [10] Zhang, Y.-n., Wu, R.-q., Schurter, H. M. & Flatau, A. B. Understanding of large auxetic properties of iron-gallium and iron-aluminum alloys. *Journal of Applied Physics* **108** (2010).
- [11] Greaves, G. N., Greer, A. L., Lakes, R. S. & Rouxel, T. Poisson’s ratio and modern materials. *Nature Materials* **10**, 823–838 (2011).
- [12] Grima, J. N., Gatt, R., Alderson, A. & Evans, K. E. On the origin of auxetic behaviour in the silicate α -cristobalite. *J. Mater. Chem.* **15**, 40–43 (2005).

- [13] Baughman, R. H., Shacklette, J. M., Zakhidov, A. A. & Stafstro, S. Negative Poisson's ratios as a common feature of cubic metals. *Nature* **392**, 362–365 (1998).
- [14] Azzopardi, K. M., Brincat, J. P., Grima, J. N. & Gatt, R. Anomalous elastic properties in stishovite. *RSC Advances* **5**, 8974–8980 (2015).
- [15] Grima, J. N. *et al.* Natrolite: A zeolite with negative Poisson's ratios. *J. Appl. Phys.* **101**, 8–11 (2007).
- [16] Goldstein, R. V., Gorodtsov, V. A. & Lisovenko, D. S. Auxetic mechanics of crystalline materials. *Mech. Solids* **45**, 529–545 (2010).
- [17] Goldstein, R. V., Gorodtsov, V. A. & Lisovenko, D. S. Classification of cubic auxetics. *Physica Status Solidi (B) Basic Research* **250**, 2038–2043 (2013).
- [18] Goldstein, R. V. & Gorodtsov, V. A. Average Poisson's ratio for crystals. Hexagonal auxetics. *Letters on Materials* **3**, 7–11 (2013).
- [19] Cardarelli, F. *Materials Handbook* (Springer, London, 2008), 2nd edn.
- [20] Lakes, R. S. Foam structures with a negative Poisson's ratio. *Science* **235**, 1038–1040 (1987).
- [21] Alderson, A. A triumph of lateral thought. *Chemistry and Industry* **10**, 384–391 (1999).

REVIEWERS' COMMENTS:

Reviewer #1 (Remarks to the Author):

I am happy with the changes that the authors have made in response to my original review and also the extensive work done on addressing the issues raised by two other reviewers. I am OK with this interesting paper now proceeding to publication.

Reviewer #2 (Remarks to the Author):

I am happy with the revisions that the authors have made following my suggestions. The revised manuscript is acceptable for publication.

Reviewer #3 (Remarks to the Author):

The authors have answered all my concerns in their reply letter, thus I recommend this manuscript for publication in Nature Communication.

Reviewer #1:

I am happy with the changes that the authors have made in response to my original review and also the extensive work done on addressing the issues raised by two other reviewers. I am OK with this interesting paper now proceeding to publication.

Thank you very much for your helpful clarification questions and comments. We appreciate the time you took to review our manuscript.

Reviewer #2:

I am happy with the revisions that the authors have made following my suggestions. The revised manuscript is acceptable for publication.

Thank you for your comments and the helpful references which you pointed out during in the review process. We appreciate the time you took to review our manuscript.

Reviewer #3

The authors have answered all my concerns in their reply letter, thus I recommend this manuscript for publication in Nature Communication.

Thank you very much for the constructive criticism that you provided during the review process. We appreciate the time you took to review our manuscript.